# Static and Dynamic Strength Indicators in Paralympic Power-Lifters with and without Spinal Cord Injury

**DOI:** 10.3390/ijerph18115907

**Published:** 2021-05-31

**Authors:** Luan José Lopes Teles, Felipe J. Aidar, Dihogo Gama de Matos, Anderson Carlos Marçal, Paulo Francisco de Almeida-Neto, Eduardo Borba Neves, Osvaldo Costa Moreira, Frederico Ribeiro Neto, Nuno Domingos Garrido, José Vilaça-Alves, Alfonso López Díaz-de-Durana, Filipe Manuel Clemente, Ian Jeffreys, Breno Guilherme de Araújo Tinoco Cabral, Victor Machado Reis

**Affiliations:** 1Graduate Program of Physical Education, Federal University of Sergipe (UFS), São Cristovão 49100-000, Brazil; luantelespersonaltrainer@gmail.com (L.J.L.T.); acmarcal@yahoo.com.br (A.C.M.); 2Group of Studies and Research of Performance, Sport, Health and Paralympic Sports (GEPEPS), Federal University of Sergipe (UFS), São Cristovão 49100-000, Brazil; dihogogmc@hotmail.com; 3Department of Physical Education, Federal University of Sergipe (UFS), São Cristovão 49100-000, Brazil; 4Graduate Program of Physiological Science, Federal University of Sergipe (UFS), São Cristovão 49100-000, Brazil; 5Cardiovascular & Physiology of Exercise Laboratory, University of Manitoba, Winnipeg, MB R3T 2N2, Canada; 6Department of Physical Education, Federal University of Rio Grande do Norte, Natal 59064-741, Brazil; paulo220911@hotmail.com (P.F.d.A.-N.); brenotcabral@gmail.com (B.G.d.A.T.C.); 7Brazilian Army Research Institute of Physical Fitness (IPCFEx), Rio de Janeiro 70630-901, Brazil; eduardoneves@utfpr.edu.br; 8Institute of Biological Sciences and Health, Federal University of Viçosa, Campus Florestal, Viçosa 35690-000, Brazil; ocostamoreira@gmail.com; 9Paralympic Sports Program, SARAH Rehabilitation Hospital Network, Brasilia 71535-005, Brazil; fredribeironeto@sarah.br; 10Research Center in Sports Sciences, Health Sciences and Human Development (CIDESD), University of Trás-os-Montes e Alto Douro, 5001-801 Vila Real, Portugal; ndgarrido@gmail.com (N.D.G.); josevilaca@utad.pt (J.V.-A.); victormachadoreis@gmail.com (V.M.R.); 11Sports Department, Physical Activity and Sports Faculty-INEF, Universidad Politécnica de Madrid, 28040 Madrid, Spain; alfonso.lopez@upm.es; 12Escola Superior Desporto e Lazer, Instituto Politécnico de Viana do Castelo, Rua Escola Industrial e Comercial de Nun’Álvares, 4900-347 Viana do Castelo, Portugal; filipe.clemente5@gmail.com; 13Instituto de Telecomunicações, Delegação da Covilhã, 1049-001 Lisboa, Portugal; 14Setanta College, E45 Thurles, Ireland; Ian.jeffreys@setantacollege.com

**Keywords:** spinal cord injury, para-athletes, muscle strength, disabled persons, athletic performance

## Abstract

Background: In Paralympic powerlifting (PP), athletes with and without spinal cord injury (SCI) compete in the same category. Athletes with SCI may be at a disadvantage in relation to the production of muscle strength and the execution of motor techniques. Objective: To analyze the indicators force, dynamic and static, at different intensities, on performance in athletes with and without SCI. Methods: The sample was composed of two groups of PP athletes: SCI (30.57 ± 4.20 years) and other deficiencies (OD; 25.67 ± 4.52 years). Athletes performed a test of maximum isometric force (MIF), time to MIF (Time), rate of force development (RFD), impulse, variability and fatigue index (FI), dynamic tests Mean Propulsive Velocity (MPV), Maximum Velocity (Vmax) and Power. Results: There were differences in the SCI in relation to OD, 50% 1RM (*p* < 0.05), in relation to MPV and Vmax. There were no differences in the static force indicators. Regarding EMG, there were differences between the SCI triceps in relation to the previous deltoid (*p* = 0.012). Conclusion: We concluded that the static and dynamic strength indicators are similar in Paralympic powerlifting athletes with spinal cord injury and other disabilities.

## 1. Introduction

Spinal cord injury (SCI) is a condition that tends to be debilitating, and annually around half a million people are affected worldwide [1]. These injuries are traumatic (car accident and falls) or non-traumatic (myelomengiocele, spinal stenosis, transverse myelitis and tumor) [1,2]. SCI usually presents physical disability and impaired quality of life in several aspects, such as physical, social and environmental [3]. Most of the people with SCI are male and under 30 years of age [4]. The forms of rehabilitation are of paramount importance, where physical exercises tend to represent a very important strategy [1]. The practice of physical and sports activities has shown great importance, not only in physical health but also in general well-being [5]. Sports practice increases the sense of belonging, promoting social interaction and emotional support [6]. As a result, encouraging sports practice, as well as participation in competitions, can be an important aspect of total rehabilitation. To facilitate this, understanding the challenges presented by people with disabilities becomes important [7].

In the context of parasports, it is necessary to take into account that SCI tends to provide secondary complications, notably in relation to the damage of the autonomic nervous system (ANS) [3]. This is because the ANS serves as a control that interferes with the regulation of many physiological functions, blood pressure (BP), heart rate (HR), respiratory rate, urination and intestinal motility, among others. In this sense, physical performance depends on a coordinated and broadly functioning ANS [8]. SCI tends to compromise athletic performance, influencing the difficulty of maintaining strength, power, velocity, endurance and specific and important neuromotor skills required for the sport. In addition, performance tends to be impaired due to premature fatigue, resulting from the interaction that involves multiple physiological systems and mechanisms [9]. Therefore, the loss or decrease in autonomic control, which tends to be impaired in people with SCI, tends to impair athletic performance across a range of potential severities, ranging from low performance caused by fatigue to serious risks, including death [10].

When assessing sports practice, Paralympic powerlifting (PP) appears to be an excellent mode of sports practice for the disabled, being a sport characterized by the manifestation of strength, and only has the bench press adapted from conventional powerlifting (CP) [11]. Men and women with physical disabilities, especially in the lower limbs, may be eligible for the PP dispute [11]. The main difference in relation to CP is that the Paralympic sport is performed on the bench press with the lower limbs on the bench, with the athletes being fixed to the bench through bands [11].

The number of athletes in the sport has increased, and the results have been increasingly prominent [12,13]. Studies have focused more on the issue of health in relation to the etiology and prevention of injuries [14], recovery methods [15], warm-up [16], or even the width of the catch in sport [17]. On the other hand, when evaluating the PP, where the legs are extended on the bench, the SCI tends to reduce the transfer of strength for lifting in the adapted bench press [18]. Additionally, in the SCI, the transfer would be more impaired, given the inability to maintain strength, power, speed, and, consequently, the performance of sport-specific neuromotor skills in relation to other disabilities [9].

It is noteworthy that in other Paralympic sports, such as swimming, athletics and bocce ball, the athletes undergo analysis carried out by health professionals; the analyses will classify the athletes functionally and allocate them in subcategories [11]. For example, in the Paralympic Bocce, there are Categories BC2 and BC3. BC2 is characterized by athletes who have cerebral palsy and who are able to move the wheelchair and perform moves without the aid of an external person or additional equipment to perform the throwing of bochas [11]. BC3 is composed of athletes who have various disabilities; however, they do not have the ability to move the wheelchair without assistance from another person, and they do not have enough muscle strength to perform the throwing of the balls. Therefore, they use external equipment to perform the launch [11].

As exemplified, the subcategories of functional classification enable a fair competition among Paralympic athletes; however, this functional classification does not occur in PP [11]. In PP, the classification is purely binary, where the subject is classified as eligible (i.e., having an injury that impairs lower limbs) or ineligible [11]. Thus, a stratified functional classification is necessary to enable a fair competition between athletes in the PP. When evaluating the PP, where the legs are extended on the bench (an adapted bench press), the transfer of force could be impaired, making it difficult to maintain strength, power and speed, with decreased neuromotor abilities [11,18]. In this sense, we raised the hypothesis that athletes with SCI would present different patterns of strength and activation in relation to other deficiencies eligible for the sport [11].

The aim of this study was to analyze mechanical, dynamic and static indicators of strength, at different intensities, on performance in athletes with Spinal Cord Injury and other deficiencies of Paralympic powerlifting.

## 2. Materials and Methods

### 2.1. Sample

The sample consisted of 19 male Paralympic powerlifting athletes: 9 with spinal cord injuries and 10 with other deficiencies (OD). The participants were classified competitors, eligible to compete in the sport [11], with at least 12 months of experience and training. Among the deficiencies in the SCI group, eight had spinal cord injury by accident and one due to injury caused by the parasite Schistosoma Mansoni in the spinal cord, all with spinal cord injury below the eighth thoracic vertebra. In the other deficiencies group (OD), four subjects suffered from amputation, three with arthrogryposis, two with lower limb disability due to traumatic brain injury and one due to nerve damage to the right lower limb. The athletes participated in the study on a voluntary basis and signed a free and informed consent form, in accordance with resolution 466/2012 of the National Research Ethics Commission (CONEP), of the National Health Council, and the ethical principles expressed in Helsinki Declaration (1964, reformulated in 2013), by the World Medical Association. This study was approved by the Research Ethics Committee of the Federal University of Sergipe, CAAE: 2.637.882 (date of approval: 7 May 2018). The sample characterization is shown in Table 1.

The sampling power was calculated a priori using the open source software G*Power^®^ (Version 3.0; Berlin, Germany), choosing a “F family statistics (ANOVA)” considering a standard α < 0.05, β = 0, 80 and the effect size of 1.33 found for the Rate of Force Development (RFD) in Paralympic powerlifting athletes in the study by Sampaio et al. [13]. Thus, it was possible to estimate a sample power of 0.80 (F _(2.0)_: 4.73) for a minimum sample of eight subjects per group, suggesting that the sample size of the present study has statistical strength to respond to the research approach.

This study followed a static and dynamic force test, we analyzed the effects of two different classifications of disabilities (i.e., SCI and OD; see Table 1) on the performance of Paralympic powerlifting athletes at the national level. The study lasted three weeks. The first week aimed at familiarization with the tests of 1 Maximum Repetition (1RM) and 72-h later with the dynamic and static tests. At week 2, the 1RM and static tests were performed with a 72-h interval. Records in these sessions included maximum isometric force (MIF), time to MIF (Time), rate of force development (RFD), impulse, variability and fatigue index (FI). Finally, in week 3, the two sessions comprised dynamic tests at 40 to 60% 1-RM and, 72-h later, at 70 to 90% 1-RM. In both sessions, measurements included mean propulsive velocity (MPV), maximum velocity (Vmax) and power and Surface Electromiography (sEMG). All tests were performed on different days at the same time (between 9:00 a.m. and 12:00 p.m.) at temperatures ranging between 23 °C and 25 °C with a relative humidity of ~60%. All tests were performed on an adapted bench press in the supine position. The study was carried out at the Federal University of Sergipe.

### 2.2. Instruments

The body mass of the athletes was measured with the subjects in a sitting position using an appropriate Michetti digital electronic scale, Model Mic Welchair (Michetti, São Paulo, SP, Brazil). An official 210 cm long straight bench and a 220 cm long 20 lg bar were used herein (Eleiko Sport AB, Halmstad, Sweden), both pieces of equipment were approved by the International Paralympic Committee (IPC) [11].

### 2.3. Determination of Load

The athletes started the testing with a self-selected load estimated to be the maximal load. Weight was then added until the maximum load was attained. If the participant overestimated the initial load, 2.5% of the load was subtracted before a new attempt [20]. A rest period of 3 to 5 min was provided between attempts, according to the participants’ perception of recovery [13,15,16]. The coefficient of variation between the two measures was at least 94%.

### 2.4. Warm Up

The participants performed a standardized warm-up for the upper limbs, using three exercises (abduction of the shoulders with dumbbells, military press with dumbbells, and medial and lateral rotation of the arm to warm up the rotator cuff with dumbbells) as described elsewhere [15], for approximately 15 min.

### 2.5. Dynamic Evaluation

The athletes were evaluated during the competitive phase of the season and were familiar with the testing procedures due to their constant training and testing routines. To measure the velocity of movement, a valid and reliable linear position transducer [21], Chronojump (Chronojump, BoscoSystem, Barcelona, Spain), was attached to the bar. The MPV and VMax were collected for analysis purposes with loads of 100% 1RM [13,16,22,23,24].

### 2.6. Isometric Force Measurements

The measures of muscle strength, RFD (N·s^−1^), MIF (N), FI (%) and time to MIF (s), were determined by a Chronojump force sensor (Chronojump, BoscoSystem, Barcelona, Spain) as described in detail elsewhere [16]. The perpendicular distance between the force sensor and the center of the joint was determined and used to calculate joint torques and FI [13,15,16,25]. MIF was measured by the maximum isometric force generated by the muscles of the upper limbs. The MIF, the FI and the RFD were calculated, as explained elsewhere [16,26].

### 2.7. Surface Electromyography

The electromyographic signals were captured on the dominant side, using double electrodes Meditrace (Tyco/Kendall, Mansfield, MA, USA), positioned parallel to the muscle fibers, 2.0 cm from the center at the point of greatest muscle area of the following muscles: brachial triceps (long head), anterior deltoid and in the sternal and clavicular portions of the pectoralis major, on both sides of the body. The ground electrode was positioned over the olecranon. The skin area where the electrodes were placed was previously shaved and cleaned with 70% alcohol solution. The electrodes (11.0 mm contact diameter and a 2.0 cm center-to-center distance) were placed along the presumed direction of the underlying muscle fiber according to the recommendations by SENIAM [27]. For data acquisition, one set was used with one repetition and a maximum load of 100% 1RM. The marker function was used to define the data intervals for each height in the sticking region. The electrodes were placed on the muscle belly along the estimated direction of the muscle fiber. Before placing the electrode, the skin was scraped, sanded and washed with alcohol according to the recommendations of SENIAM [27]. The electrodes were placed in four locations: the pectoral clavicular portion (~4 cm medial to the axillary crease, in the second intercostal space under the midpoint of the clavicle), the pectoral scapular portion (~6 cm medial to the axillary crease, and between the third and fourth intercostal space under the point proximal to the Sternum), anterior deltoid (1.5 cm distal and anterior to the acromion) and brachial triceps (long head, ~3 cm medial and 50% on the line between the acromion and the olecranon) [28].

The equipment used was an electromyographic MIOTEC^®^ (MIOTEC, Porto Alegre, RS, Brasil), with eight channels. The data were filtered (second-order Butterworth band-pass filter of 20–500 Hz; notch of 60 Hz). The signal amplitude was calculated through the mean square root (MSR), which was normalized by the percentage of the maximum voluntary isometric contraction (MVIC). MVIC acquisition occurred before the test was performed, and a lift was carried out that remained in an isometric state for 5 s. CVMI values were recorded by the equipment and used for normalization. The equipment program issues a report with the values after normalization that were used for analysis in this study, adapted from Golas et al. [29].

### 2.8. Statistics

Descriptive statistics were performed using measures of central tendency, mean (X) ± Standard Deviation (SD) and 95% confidence interval (95% CI). To verify the normality of the variables, the Shapiro–Wilk test was used. The data for all variables were homogeneous and normally distributed. To compare the conditions of exercise and moments of measurement (40% × 50% × 60% × 70% × 80% × 90% of 1RM), the ANOVA (Two Way) test was performed with Bonferroni’s Post Hoc. To check the effect size, the partial Eta squared (η2p) was used, adopting values of low effect (≤0.05), medium effect (0.05 to 0.25), high effect (0.25 to 0.50) and very high effect (>0.50) [30]. In comparisons between groups (SCI × OD), a Student’s t-test was used. For the t-test, an effect size (Cohen’s d) was calculated, adopting values of low effect (≤0.20), medium effect (0.20 to 0.80), high effect (0.80 to 1.20) and very high effect (>1.20) [31,32]. The variation coefficient (CV%) was calculated by the formula: CV% = (standard deviation (SD)/mean) × 100. In addition, we calculated the intraclass correlation coefficient (ICC), whose magnitudes were determined as [29]: absence: <0; bad: 0–0.19; weak: 0.20–0.39; moderate: 0.30–0.59; substantial: 0.60–0.79; almost complete: ≥0.80. Statistical analyses were performed using the Statistical Package for the Social Science (SPSS) version 22.0 software (IBM, North Castle, New York, NY, USA). The level of significance was set at *p* < 0.05.

## 3. Results

The presented results were found in MPV (m·s^−1^; Figure 1) and Vmax (m·s^−1^; Figure 2) in subjects SCI and OD, in the percentages of 40% to 90% of 1 RM.

Mean Propulsive Velocity (m·s^−1^) measured from 40% to 90% of 1RM in SCI and OD subjects. a: Indicates difference in SCI between 40% in relation to 70% (*p* = 0.002), 80% (*p* = 0.005) and 90% 1RM (*p* < 0.001); b: Indicates differences in SCI between 50% in relation to 70 and 90% (*p* < 0.001) and 80% 1RM (*p* < 0.024); c: Indicates differences in SCI between 60% in relation and 90% 1RM (*p* = 0.014); d: Indicates differences in SCI between 40% compared to 60% (*p* = 0.025), 70% (*p* = 0.008) and 90% 1RM (*p* = 0.002); e: Indicates differences in OD between 50% in relation to 70% (*p* = 0.007), 80% (*p* = 0.002) and 90% of 1RM (*p* < 0.001); f: Indicates difference in OD between 60% in relation to 80% (*p* = 0.001) and 90% of 1RM (*p* < 0.001); g: Indicates differences in OD between 80% in relation to 90% 1RM (*p* = 0.004); #: Indicates differences in the SCI in relation to OD 50% 1RM (*p* = 0.003). The effect was very high intra group η2p = 0.936 (very high effect), and inter group, small effect η2p = 0.173 (Medium effect).

Maximum Velocity (m s^−1^) measured from 40 to 90% of 1RM in SCI and OD subjects. a: Indicates difference in SCI between 40% in relation to 70% (*p* = 0.006) and 90% 1RM (*p* = 0.002). b: Indicates differences in the SCI between 50% in relation to 70% (*p* = 0.001) and 90% 1RM (*p* = 0.002). c: Indicates differences in SCI between 60% in relation and 90% 1RM (*p* = 0.023). d: Indicates differences in OD between 40% in relation to 70% (*p* = 0.008), 70% (*p* = 0.016) and 90% 1RM (*p* = 0.003). e: Indicates differences in OD between 50% in relation to 70% (*p* = 0.016), 80% (*p* = 0.022) and 90% of 1RM (*p* = 0.004). f: Indicates difference in OD between 60% in relation to 80% (*p* = 0.011) and 90% of 1RM (*p* = 0.002). g: Indicates differences in OD between 80% in relation to 90% 1RM (*p* = 0.001). #: Indicates differences in SCI in relation to OD 50% 1RM (*p* = 0.049). The effect was very high intra group η2p = 0.910 (very high effect), and inter group, small effect η2p = 0.177 (Medium effect).

The results found in the Power (W) of the subjects SCI and OD, in the percentages of 40% to 90% of 1 RM, are shown in Figure 3.

Power (W) measured from 40% to 90% of 1RM in subjects SCI and OD; a: Indicates differences in OD 60% compared to 90% 1RM (*p* = 0.011); b: Indicates differences in OD 80% compared to 90% 1RM (*p* = 0.008); The effect was very medium intra Group η2p = 0.529 (very high effect), and inter Group small effect η2p = 0.144 (medium effect).

The results found in the dynamic mechanical variables (VMP, Vmax, Pot and 1RM) and isometric (FIM, Time, RFD, Impulse, Variability, FI) of the subjects SCI and OD are shown in Table 2.

The results found in the surface electromyography at the intensities of 40% to 90% of 1 RM, are shown in Table 3. There was greater activation of the triceps in the OD group in relation to the anterior deltoid muscle of the SCI, indicating a pattern of muscular activation differentiated between the groups.

## 4. Discussion

The aim of this study was to analyze mechanical, dynamic and static indicators of performance in Paralympic powerlifting athletes with and without spinal cord injury. The main findings were: (i) For intragroup propulsive velocity, the higher the 1RM load, the lower the propulsive velocity; (ii) In the comparison between propulsive velocity in both groups, there was a 50% difference between SCI and OD; (iii) There was a difference in the maximum velocity in relation to the 40% to 90% percentages of 1RM intra groups for SCI and OD; (iv) There was a difference in power for the OD at 60% of 1RM compared to 90% of 1RM; (v) There was a difference in the activation of the triceps brachii in the SCI group in relation to the activation of the anterior deltoid in the OD.

The present study found that the greater the load of 1RM, the lower the propulsive velocity of athletes of Paralympic powerlifting. This is to be expected and is in line with the force–velocity relationship. García-Ramos et al. [23] found that the bench press postures (i.e., arched or flat) can influence the velocity in powerlifting. Elliott et al. [33] concluded that a more arched posture allows a more vertical displacement of the bar. It can lead to an improvement in the force exerted in terms of transfer [34]. Given these perspectives, athletes with SCI can be harmed, since they are unable to make a good bow, and due to injury, they point out difficulties in transferring the strength from the lower limbs to the upper limbs. In view of the inability to maintain the position, in addition to the damage to strength, power, velocity, and, consequently, the performance of sport-specific neuromotor skills in relation to athletes who have other disabilities [9].

In the specific case of the sample herein, Paralympic powerlifters are unable to move their lower limbs and depend on their upper limbs for their activities, such as pushing wheelchairs, among others. Theisen [35] found that these restrictions result in interfering with performance, as upper limb exercise causes a reduced cardiovascular response when compared to lower limb exercise. Theisen [34] also found that when cycling with upper limbs, the maximum power and VO2peak of people without SCI were reduced by approximately 40% and 25%, respectively, when compared to cycling with lower limbs. On the other hand, a study that compared neuromuscular response when bench pressing at loads of 60% to 100% 1RM found differences in the anterior deltoid, triceps, and pectoralis major activation between disabled and non-disabled athletes [36]. Thus, in general, with the change in support, especially with athletes with greater disabilities, higher loads tend to promote an increase in muscle activity [37], and these changes tend to be related to muscle control [38]. This could help to explain the different activation herein between SCI and OD athletes. Moreover, when assessing whether there would be differences between the normal position vs. with the chest arched, it was found that there were no significant differences in total load, bar trajectory and average bar speed; indicating that greater support, as allowed in competitions with the use of a belt, may enable a greater stability for athletes [39].

In Paralympic powerlifting, where the legs must be extended on the bench, this position tends to reduce the transfer of force for lifting in the adapted bench press [18]. This fact was not observed in the present study, with similar results in both groups recorded. However, in the present study, when maximum velocity was analyzed, a difference was found only within the group. On the other hand, higher velocities were observed for less trained subjects than for more trained ones [22], where the velocity in more trained subjects was lower than in less trained subjects. Loturco et al. [24] found, in a study with Paralympic athletes, that certain segments had extremely low execution velocity in the adapted bench press. However, our study found no differences between groups. Perhaps this is explained by the adaptations not mentioned in other studies, where athletes with disabilities tend to produce better performance in terms of the production of strength in the upper limbs [36,37,38]. This fact needs to be better explored, but it may have some relation as to the fact that they use the upper limbs as a form of locomotion when using wheelchairs or crutches [40].

Regarding power, differences were found only in OD at an intensity of 90% in relation to 60% and 80% 1RM. There is no difference in SCI and between groups. It is possible that diaphragm fatigue induced by exercise is the justification in relation to the power in the spinal cord injury group. The diaphragm contracts and expands the rib cage during inspiration, at the same time that it opposes the mechanical forces transmitted by the thorax. This added to the lying position in dorsal decubitus could affect the breathing dynamics and hinder the manifestation of force in spinal cord injury [41,42].

In the dynamic and isometric mechanical variables, there were no differences between the groups analyzed by the present research. This may be due to the fact that Paralympic powerlifting athletes are better able to apply force against heavier loads and, consequently, at lower velocities, this would be a consequence of training with higher loads, which tends to generate specific adaptations [43]. Therefore, this could also justify the non-statistical differences between the groups in the parameters of static strength since this would be a constant in training aimed at maximum strength, in this case, the Paralympic powerlifting [44].

Regarding EMG, greater activation was observed between the triceps in the SCI in relation to the anterior deltoid of the OD. In this sense, changes in EMG activity between able-bodied and disabled athletes during bench press movements are most likely linked to tonic muscle function. Brennecke et al. [45] recommend alternate concepts of muscle stimulation during the same and incremental loads that have not yet been specifically clarified. The first theory applies to low energy consumption. The second principle is based on the interpretation of external forces, such as gravity, while the third principle consists of muscle synergy between different parts of the body. In the case of a low limb disability athlete with restricted kinesthetic awareness and proprioception in this region, deficits can be offset by improved coordination of the motor units and increased engagement of specific muscle groups during the bench press.

The SCI athletes had a larger training experience compared with OD. However, the level of performance, as shown by the absolute and relative bench press 1-RM, was not significantly higher. Moreover, a study that aimed to compare the reliability and magnitude of the speed variables between three variants of the bench press exercise in individuals with and without training experience, concluded that regardless of the type of bench press variant, no significant differences in execution speed were observed between experienced and non-experienced participants [46]. Therefore, we may suggest that the larger experience in SCI group did not play a significant role herein.

However, despite the relevance of the results, the present study has a limitation that the small sample size limits the generalizability of these results. Larger trials utilizing a large sample are required to enhance the applicability of these findings. Another limitation of the study was that it comprised only male athletes of national level. In addition, the evaluations were performed only with the adapted bench press and cannot be extrapolated to other muscle groups or other movements related to activities of daily living. In addition, there was a difference in training experience between the two groups herein.

It is suggested that further studies be carried out to verify the effect of training time on elite athletes. Other studies could also evaluate the activities of daily living and relating other muscle groups in PP athletes.

## 5. Conclusions

It was concluded that the indicators of static and dynamic strength for the bench press Paralympic powerlifting are similar for athletes with SCI and with other disabilities. It is suggested that sports training may supply part of the expected loss of strength in subjects with SCI.

In view of the results herein, we endorse the current rules of functional classification, with a single classification for Paralympic powerlifting (eligible or not eligible).

Finally, coaches can give more emphasis to the muscles most demanded according to the deficiency, where the triceps tend to be more activated in the SCI and the deltoids in the OD condition. In addition, it appears that athletes with SCI tend to produce more speed and power at higher loads (i.e., 80% of 1RM), and this should be accounted for when using movement speed to control training load.

## Figures and Tables

**Figure 1 ijerph-18-05907-f001:**
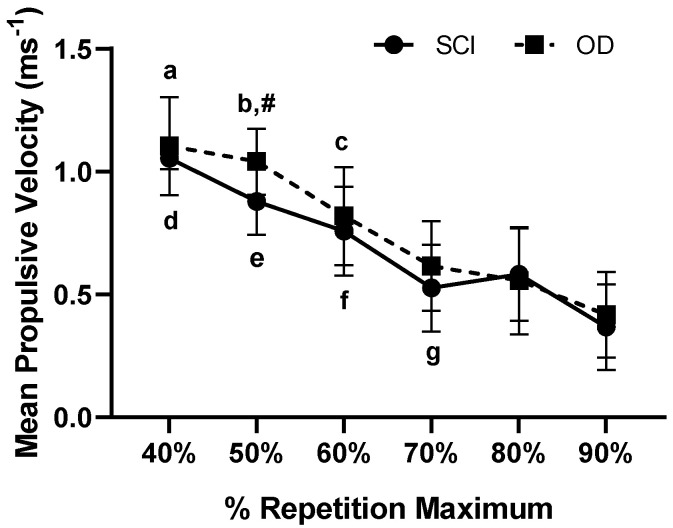
Analysis of dynamic force indicators, mean propulsive velocity (m s^−1^) measured from 40% to 90% of 1RM in sLM and OD groups. SCI: Spinal Cord Injury; OD: Other Disability.

**Figure 2 ijerph-18-05907-f002:**
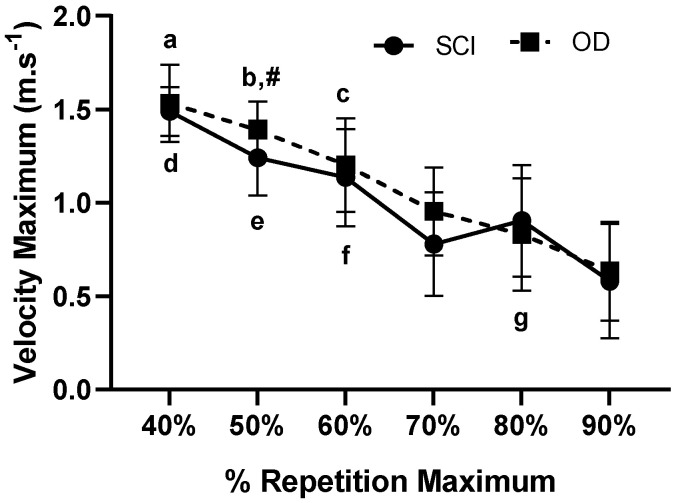
Analysis of dynamic force indicators maximum velocity (m·s^−1^) measured from 40% to 90% of 1RM in LM and OD groups. SCI: Spinal Cord Injury; OD: Other Disability.

**Figure 3 ijerph-18-05907-f003:**
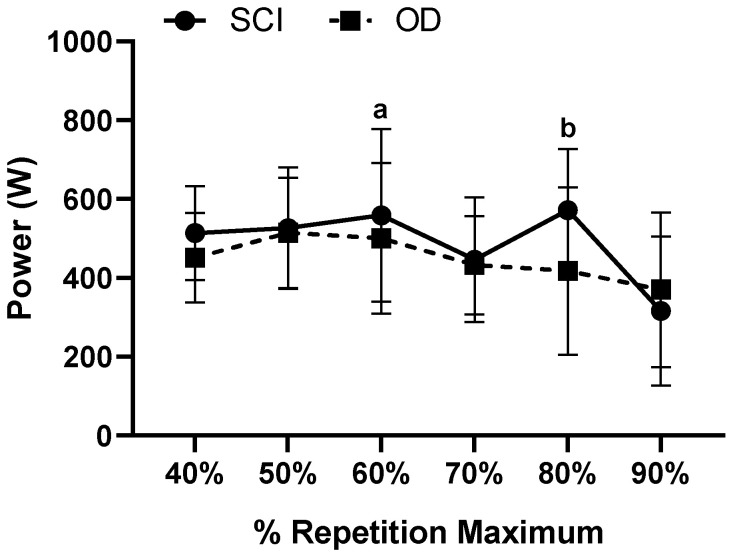
Power (W) measured from 40% to 90% of 1RM in subjects LM and OD. SCI: Spinal Cord Injury; OD: Other Disability.

**Table 1 ijerph-18-05907-t001:** Sample characterization.

Characteristics	Spinal Cord Injury	Other Deficiencies	*p*	ICC	CV	α
Age (years)	30.57 ± 4.20	25.67 ± 4.52	0.232	0.308	5.04	0.302
Body mass (kg)	81.29 ± 21.68	73.89 ± 17.56	0.400	0.375	0.16	0.371
Experience (years)	3.07 ± 0.82	2.23 ± 0.86	0.023 ^#^	0.095	15.23	0.151
1RM bench press test (kg)	122.29 ± 25.88 *	106.40 ± 31.17	0.701	0.251	10.19	0.278
1RM/weight	1.54 ± 0.32 **	1.48 ± 0.37 **	0.701	0.308	6.77	0.302

^#^*p* < 0.05 (independent “t” test). * All athletes with loads that keep them in the top 10 of their categories nationwide. ** Values above 1.4 in the bench press would be considered elite athletes, according to Ball and Weidman [19]. ICC: Intraclass Correlation Coefficient; CV: Variation Coefficient.

**Table 2 ijerph-18-05907-t002:** Indicators of dynamic and isometric force with 100% of 1RM (mean ± standard deviation) in spinal cord injured and other disabled individuals.

Force Indicators	SCI	OD	*p*	Cohen’s d
MPV (m·s^−1^)	0.17 ± 0.13	0.20 ± 0.08	0.668	0.33 ^b^
Vmax (m·s^−1^)	0.32 ± 0.19	0.34 ± 0.08	0.742	0.39 ^b^
Power (w)	178.86 ± 111.10	194.89 ± 82.16	0.952	0.86 ^c^
1RM (kg)	122.29 ± 25.88	106.44 ± 31.17	0.179	0.96 ^c^
MIF (N)	867.84 ± 172.08	856.74 ± 167.84	0.945	0.24 ^b^
Time (µs)	2651.73 ± 1478.78	2359.73 ± 1338.24	0.601	0.75 ^b^
RFD (N·s^−1^)	2362.11 ± 1078.38	2397.57 ± 867.79	0.561	0.05 ^a^
Impulse (N·s)	4011.52 ± 815.44	3828.39 ± 819.33	0.776	0.44 ^b^
Variability (N)	44.12 ± 26.44	37.37 ± 11.83	0.308	0.34 ^b^
FI (%)	8.04 ± 2.44	10.84 ± 5.59	0.213	0.67 ^b^

^a^: Small Effect (≤0.20), ^b^: Medium Effect (0.20 to 0.80), ^c^: High Effect (0.80 to 1.20), ^d^: Very High Effect (>1.20); MPV: Mean Propulsive Velocity; Vmax: Maximum Velocity; 1RM: 1 Repetition Maximum, MIF: Maximum Isometric Force; Time: Time to MIF; RFD: Rate of Force Development; FI: fatigue index, SCI: Spinal cord injury; OD: Other Deficiencies.

**Table 3 ijerph-18-05907-t003:** Surface Electromyography in the different muscle groups in SCI and OD subjects (X ± SD and 95% CI).

Group	Pectoral Sternal (X ± SD) 95% CI	Pectoral Clavicular (X ± SD) 95% CI	Deltoide Anterior (X ± SD) 95% CI	Triceps (X ± SD) 95% CI	*p*	η2p
SCI	115.41 ± 104.57 27.99 − 202.83	95.81 ± 52.48 51.94 − 139.69	34.03 ± 23.38 * 14.48 − 53.58	109.56 ± 43.53 * 73.16 − 145.95	0.012	0.368 ^b^
OD	141.80 ± 108.49 51.10 − 232.50	152.45 ± 86.94 33.83 − 308.73	121.96 ± 113.45 27.11 − 216.81	240.56 ± 166.75 101.15 − 379.97	0.116	0.241 ^a^

* *p* < 0.05 (two-way ANOVA, and Bonferroni’s Post Hoc). ^a^: medium effect (0.05 to 0.25), and ^b^: high effect (0.25 to 0.50); SCI: Spinal cord injury; OD: Other Deficiencies; mean (X) ± Standard Deviation (SD) and 95% confidence interval (95% CI).

## Data Availability

The data used in the study can be obtained from Group of Studies and Research of Performance, Sport, Health and Paralympic Sports (GEPEPS), Federal University of Sergipe (UFS), São Cristovão, Sergipe 49100-000, Brazil; fjaidar@academico.ufs.br (F.J.A.).

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
