# Peer review of "Static and Dynamic Strength Indicators in Paralympic Power-Lifters with and without Spinal Cord Injury"

_ijerph, 2021, doi:10.3390/ijerph18115907_

Round 1

Reviewer 1 Report

Interesting subject matter presented very clearly and in detail.  Although I do not have extensive knowledge of the sport, it appears the authors have conducted a thorough and appropriate analysis of relevant strength factors.

I appreciate the goal to possibly creater further subcategories in PP if there is indeed a disparity for certain impairments. Is it possible that athletes with SCI impairments might be self-selecting to not compete in powerlifting due to having greater impairment compared to other para-athletes? If that is the case, the athletes studied here may not represent equivalent percentages in their categories (SCI vs OD). Are the authors able to comment on the percentage of athletes with SCI competing in other events as compared to the percentage competing in PP?

In the conclusion, the authors should return to the topic of functional classification.  Do they agree that the binary classification fits? Or do they have enough evidence to request that further inquiries be made?

A few comments about the figures and table:

  • I like the statistical results presented as charts, but recommend splitting Figure 2 into two separate figures so that the caption is easier to read.
  • Shouldn't the values in Table 2 be presented with periods instead of commas? It would better match the style of the body text.
  • Comment about the *p in Figure 4's caption and also change the comma to a period

Minor typos/ stlye comments:

  • p3 #107 missing citation
  • p3 #119 missing ) after (OD
  • p7 #249-255. Use either commas or semi-colons between each element in a list
  • p7 #258-271 ie should be i.e. Also, remove commas surrounding the citations for [23] and [32] and after [34]. Throughout the article, check for extra commas around the citations.

Author Response

Thanks for the considerations, the responses to the review as per the attached document.

Reviewer 2 Report

Although this study is very intriguing in general, I have a few comments regarding some issues:

General comments

1# Table.1 - "Sample characterization" there is no statistical analysis between the two groups: size, weight, years of experience, 1RM bench press test and 1RM/weight. In particular, significant statistical differences between groups in size, years of experience, age and weight may influence the main results.

2# there is no hypothesis in the introduction.

3#Why was the 6-month period of training adopted as the inclusion criteria? (L120-121)

4# L160 - "A 3.0 to 5.0 min rest was provided between attempts" why was there a different time of rest? Whose methodology was it compatible with?

5# The description of electromyography examination raises many questions:

  • L182/202 - The authors wrote  "Electromyography (EMG)" but in my opinion the description concerns the use of surface electromyography (sEMG). What type of electromyogram was used?
  • L188 – What is the percent of alcohol that was used?
  • L194/195 - When was the EMG / sEMG survey conducted – in the morning or in the afternoon?
  • in what position was the EMG analysis carried out? According to whose methodology?

6#Please provide the answer, how the sampling power was calculated, what values ​​were adopted, the description of L134 raises reservations. What numerical values ​​were taken from the previous research?

7# The graphic form of the results is not clear to the recipient. It suggests changing to the form of tables.

8# References - incorrect description of many items (for example: 4,6,12,23,32,38). The authors should check all the items in the bibliography description.

https://www.mdpi.com/journal/ijerph/instructions

Specific comments

L51 - key words: "Paralympic Powerlifting", " Disabilities", "Performance" are not according with The Medical Subject Headings (MeSH)

L101 - "throwing. bochas" - remove the dot

L107 - citation "[xxx]" is incorrect

L119 - "(OD." change to "(OD)."

L129/130 - " (1964, reformulated in 1975, 1983, 1989, 1996, 2000, 2008 and 2013) " it is redundant information. The Helsinki Declaration as amended by the 64th WMA General Assembly, Fortaleza, Brazil, October 2013 applies

L132 - "(date of approval: May 7, 2018)" this information is not required

L133/231 "table" hang on "Table" according to ijerph-template

L173 -"[13,16,22,23,24]" change on " [13,16,22-24]"

L189 - Add the manufacturer and the name of the electrodes used.

L191 - quote [27] is wrong, change to

Hermens, H.J.; Freriks, B.; Disselhorst-Klug, C.; Rau, G. Development of Recommendations for SEMG Sensors and Sensor Placement Procedures. J. Electromyogr. Kinesiol. 2000, 10, 361–374

L209 -  add lost bracket

L325 - remove "Please add"

L334 -  remove quotation marks

Author Response

(The authors gave the same response as above.)

Reviewer 3 Report

Thank you to the authors for submitting their manuscript to International Journal of Environmental Research and Public Health; I enjoyed reading it.

There some suggestions that I think will provide clarity to the reader (outlined below).

Good luck with your amendments and I look forward to seeing the revised version.

SPECIFIC COMMENTS

It is necessary to reduce the high number of acronyms throughout the text to make easier to read it.

Abstract:

You used acronyms that you did not use them again. Review the abstract.

Introduction:

Please clarify the idea of this phrase. (line 58)

What does it mean twice the same words (lower limbs)? (line 85)

Delete “.” after “throwing”. (line 101)

What does it mean (xxx)? (line 107)

Material and Methods:

Please add the gender of study population. (line 118)

Please close the parenthesis (line 119).

Why 6 months of experience and training in sport? Maybe, it would be necessary more time to consider if dynamic and static indicators of strength, at different intensities affect the performance of study population. (line 120).

Change “eighth” to “eight” (line 123)

Could you report the reliability (i.e. ICC and CV) data of performance different tests used in your study?

Please add time of warm up. (line 163)

In table 2, change “,” to “.” (line 232)

Discussion:

What this study contributes to coaches and physical trainers? Add it thought the text.

Are there other studies to discuss in this paragraph? You spoke twice about the same study. (line 267-274)

Why did not found differences between groups in your study? Discuss it. (line 283)

These terms “ODG and SCIG” need to be described thought the text.

Please add more limitations like only one country, only one sex? (line 310-312)

Please add future lines of future.

Conclusions

What this study contributes to coaches and physical trainers? Add it.

Author Response

(The authors gave the same response as above.)

Round 2

Reviewer 2 Report

The number of corrections made by the authors is significant. However, the manuscript still requires working on it.

General comments

1#Table 1. The category “experience (years)” was highlighted as statistically significant between groups. This lowers the quality of the results obtained by the authors. Please, describe in the form of discussion, how different experiences might affect the outcome.

Furthermore, I suggest adding information about this difference in the Limitations of the study (L381-386).

2# I did not understand properly the answer to my question (3# Why was the 6-month period of training adopted as the inclusion criteria?). The authors changed the period to 12 months. CCould you please specify who was in charge of choosing this period of time? Please, add quote.

3# References - I clearly asked for a correction in the last review (general coments #8). However, the authors did not correct all items. For example:

  • item 15 - change " 17(14), 5157." for " 17(14): 5157".
  • item 17 - remove "Sep"
  • item 18 - remove "Apr"
  • item 20 - Where is the information about pages that you quote?
  • item 23 - remove "(2018)"
  • item 33 - remove "(1989)"
  • item 41 and no. 42 – a mistake in the position of the year
  • item 44 - Why did you mention both years: 1985 and 1988? Which one is correct?

https://www.mdpi.com/journal/ijerph/instructions

"Journal Articles:
1. Author 1, A.B.; Author 2, C.D. Title of the article. Abbreviated Journal Name Year, Volume, page range.

Books and Book Chapters:
2. Author 1, A.; Author 2, B. Book Title, 3rd ed.; Publisher: Publisher Location, Country, Year; pp. 154–196."

The authors should check all the items in the bibliography description.

Specific comments

L108-114 - You begin three sentences in a row with the word "Thus,". You should remove the repetitions.

L206 - add quote [27] after "recommendations of SENIAM"

L207 - change " Clavicula" for "clavicula"

L192 - remove "(sEMG)" - you have  already explained the abbreviation in L155.

L165 - remove "(1RM)", you have already explained the abbreviation in L149.

L121 / L123 use SCI or spinal cord injury, do not use both in the same sentence.

L137 - "RDF" - explain the shortcut, you use it for the first time here; remove extension in 151.

L297 - Table 3 - remove "(sEMG)".

L218 - "CVMI" – please explain what does it mean.

Author Response

I am grateful for the considerations, the corrections made in the manuscript are highlighted in blue, and the answers are attached.
